# Overcoming the Low-Stability Bottleneck in the Clinical Translation of Liposomal Pressurized Metered-Dose Inhalers: A Shell Stabilization Strategy Inspired by Biomineralization

**DOI:** 10.3390/ijms25063261

**Published:** 2024-03-13

**Authors:** Yeqi Huang, Ziyao Chang, Yue Gao, Chuanyu Ren, Yuxin Lin, Xuejuan Zhang, Chuanbin Wu, Xin Pan, Zhengwei Huang

**Affiliations:** 1College of Pharmacy, Jinan University, Guangzhou 511443, China; yeqihuang@stu.jnu.edu.cn (Y.H.); gaoyue122612@stu.jnu.edu.cn (Y.G.); rcy1094243897@163.com (C.R.); lyxinjnu@stu2022.jnu.edu.cn (Y.L.); chuanbin_wu@126.com (C.W.); 2School of Pharmaceutical Sciences, Sun Yat-Sen University, Guangzhou 510006, China; changzy@mail2.sysu.edu.cn (Z.C.); panxin2@mail.sysu.edu.cn (X.P.)

**Keywords:** liposome, metered-dose inhalers, biomineralization, biomimetic materials, clinical translation, shell structure

## Abstract

Currently, several types of inhalable liposomes have been developed. Among them, liposomal pressurized metered-dose inhalers (pMDIs) have gained much attention due to their cost-effectiveness, patient compliance, and accurate dosages. However, the clinical application of liposomal pMDIs has been hindered by the low stability, i.e., the tendency of the aggregation of the liposome lipid bilayer in hydrophobic propellant medium and brittleness under high mechanical forces. Biomineralization is an evolutionary mechanism that organisms use to resist harsh external environments in nature, providing mechanical support and protection effects. Inspired by such a concept, this paper proposes a shell stabilization strategy (SSS) to solve the problem of the low stability of liposomal pMDIs. Depending on the shell material used, the SSS can be classified into biomineralization (biomineralized using calcium, silicon, manganese, titanium, gadolinium, etc.) biomineralization-like (composite with protein), and layer-by-layer (LbL) assembly (multiple shells structured with diverse materials). This work evaluated the potential of this strategy by reviewing studies on the formation of shells deposited on liposomes or similar structures. It also covered useful synthesis strategies and active molecules/functional groups for modification. We aimed to put forward new insights to promote the stability of liposomal pMDIs and shed some light on the clinical translation of relevant products.

## 1. Introduction

### 1.1. Liposomes Act as Superior Drug Delivery Systems

Nanoparticles are highly dispersed supramolecular structures, typically consisting of polymers, with submicron dimensions that are preferably less than 500 nm [1]. Nanoscale materials have wide applications in various fields, including chemistry [2], biology [3], environment [4], and medicine [5], because they possess physical and chemical properties superior to those of bulk materials. Nanoparticles have become a promising drug delivery platform, due to their abilities to improve the stability and solubility of encapsulated goods, promote transmembrane transport, prolong circulation time, enhance safety and efficacy, and overcome the challenges of traditional delivery with untargeted biological distribution [6]. Many nanoparticle-based medicines, or nanomedicines, have been approved by the FDA, like inorganic, polymeric, and lipid-based medicines (Table 1) [7]. The global nanomedicine market was estimated at USD 53 billion in 2009 and was expected to reach USD 100 billion with a booming growth rate of 13.5% [8]. 

Lipid-based nanoparticles are the most commonly used FDA-approved nanomedicines [9], as shown in Table 1. Outstandingly, liposomes are one of the most widely used lipid-based nanoparticles as a drug delivery platform [10,11].

Liposomes are micro–nano systems composed of phospholipid and sterol, constructing one or more concentric circular bilayers [12]. The lipids self-assemble by bringing their polar head groups toward the aqueous phase and positioning their hydrophobic parts in opposite directions into a double layer, forming a closed vesicle with an aqueous core and a lipid bilayer as a wall [13]. The unique structure of liposomes enables them to effectively encapsulate hydrophilic, hydrophobic, and amphiphilic molecules or even smaller nanoparticles. Lipophilic drugs can be encapsulated in the lipid bilayer or adsorbed on the surfaces of liposomes, due to hydrophobic interactions, while hydrophilic drugs can be encapsulated in the aqueous interior of the vesicles [14] (Figure 1).

As a promising carrier, liposomes can contribute to a sustained release of the cargo drug and an improved therapeutic index due to their exceptional targeted delivery, rapid cellular uptake, biodegradability, and potential functionalization [15,16]. After years of research and development, several liposome-based products have been approved, including “star products” Doxil (doxorubicin hydrochloride liposome injection), DepoDur (morphine sulfate sustained-release liposome injection), and AmBisome (amphotericin B liposome dry powder for injection) (Table 2) [17].

According to Table 2, liposomes are currently, predominantly utilized in clinical analgesic, anti-fungal therapy, anti-bacterial therapy, and anti-tumor therapy. Liposomes hold immense potential for the treatment of diverse diseases.

### 1.2. Realistic Needs and Advantages of Liposome Pulmonary Delivery

It is established that liposomes can serve as versatile drug delivery systems for the management of multiple diseases. Currently, the threat of respiratory diseases to global public health cannot be ignored. Approximately 4 million people worldwide die from chronic respiratory diseases, which exhibits extremely high morbidity and mortality [18,19,20]. Of the five major respiratory diseases classified by the International Respiratory Society Forum, chronic obstructive pulmonary disease (COPD) affects approximately 65 million individuals globally and results in 3 million deaths annually, making it the third leading cause of death worldwide. Pneumonia and tuberculosis are among the most common and lethal infectious diseases, causing millions of deaths each year. Around 14% of children worldwide have asthma [21,22,23]. Lung cancer has a five-year survival rate of less than 19% [24]. The clinical demand to treat respiratory diseases is quite urgent globally. In response to this demand, pharmaceutical scientists began exploring the feasibility of using liposomes for the pulmonary delivery of corresponding therapeutic agents.

For the well-known liposome products listed in Table 2, the most common route of administration is injection, either intramuscularly or subcutaneously [25,26]. However, this method of drug delivery has some drawbacks, such as the invasive needle puncture being risky to occupational exposure for medical personnel and low patient compliance during long-term treatment [27]. Pulmonary administration is a non-invasive route that can improve patient compliance. The large surface area of alveolar epithelial cells (>100 m^2^) [28] enables rapid absorption by the target tissue, leading to fast action. Additionally, this route delivers the therapeutics directly to the respiratory lesion site, bypassing the first-pass effect in the liver and intestine [29]. The lower enzyme activity in the respiratory tract allows drug accumulation without the need for large doses, resulting in fewer side effects and safer treatment [30]. Noticeably, phospholipids, the main components of liposomes, are also the major ingredients of endogenous pulmonary surfactants. Therefore, liposome pulmonary delivery will possess high biocompatibility and a long retention time compared to the lung region [31]. These advantages are significant over injection administration.

Based on the mentioned advantages, in recent years, a large number of fundamental studies and clinical trials regarding liposomes for pulmonary delivery have been performed [32], elucidating that it is a burgeoning field.

## 2. Pressurized Metered-Dose Inhalers Are Promising for Liposome Pulmonary Delivery

The prerequisite for liposome pulmonary delivery is to convert the liposome system into an inhalation preparation for clinical use. As for the inhalers adapted for the inhalation preparations, nebulizers [33], Dry Powder Inhalers (DPIs) [34], Soft Mist Inhalers (SMIs) [35], and pressurized metered-dose inhalers (pMDIs) [36] can be selected. The adaptability of these inhalers with liposomes for inhalation is scrutinized below.

In terms of nebulizers, during the process of nebulizing the liposome dispersion into inhalable aerosol droplets, physical shear force may be exerted on the double layer of the liposomes [37,38]. This can result in the initial cargo loss and a reduction in delivery efficiency. Furthermore, it has a long administration time and low patient compliance [39]. For a DPI, an active inhalation device, the accuracy and reproducibility of the drug delivery dose heavily rely on the inspiratory flow rate of patients [40]. However, controlling the inspiratory flow rate is challenging under different conditions, especially in pediatric and geriatric scenarios. Additionally, patients with chronic respiratory diseases often have an insufficient lung inhalation flow, leading to a failure in achieving pulmonary deposition [41]. For SMIs, the relevant technologies are less mature, viz., the knowledge to design and prepare powders with appropriate properties have been mastered less [42]. SMI powders with unsuitable properties may cause particle aggregation due to interparticle cohesion, such as van der Waals forces. The inclusion of excipients may add to the aggregation of powders, resulting in inadequate dispersibility. Additionally, the cost of an SMI is quite high, which can impose a significant economic burden on patients [43]. 

In comparison, pMDIs may be the most adaptable inhaler for liposome pulmonary delivery. The pMDI system is composed of a drug formulation, a propellant, and a pressure-tempered vessel (Figure 2). Specifically, the drug formulation can be drug-containing suspensions, emulsions, or solutions [44]. The structural departments of traditional pressure-tempered vessels include tanks, metering valves, actuators, and interface tubes [45]. The drug formulation is delivered to the patient through a propellant ejection under actuation [46].

Compared to other inhalation devices, pressure-tempered containers for pMDI can isolate the drug formulation from the detrimental factors in the external environment (including temperature, pH, and osmotic pressure [47]), thereby improving the drug stability [48]. The metering valve can deliver the drug formulation quantitatively, with rapid onset and precise positioning, improving treatment efficacy and preventing adverse side effects [49]. The pMDI is a simple and easy-to-carry handheld inhaler with high patient compliance; it has a low cost, reducing the patient’s economic burden [50,51]. Regarding pulmonary deposition, a pMDI has a better lung deposition and emission dose than a DPI. 

Based on the advantages of a pMDI and the disadvantages of a nebulizer, DPI, and SMI, we can conclude that the pMDI may be the most adaptable inhaler for liposome pulmonary delivery. This idea poses an insightful pathway toward the clinical translation of inhalable liposomes.

## 3. Poor Stability Hampers Liposomal pMDIs from Clinical Translation

Although a liposomal pMDI has great potential for clinical applications, no products in this category have reached the market or completed clinical trials. Currently, the only similar formulation product that has achieved clinical conversion is Arikayce (amikacin liposome inhalation suspension), which is a nebulizer-based formulation [52]. Nonetheless, this product can provide little reference value for the clinical translation of liposomal pMDIs, as the formulation designs of pMDIs and nebulizers hardly share common ground.

We perceive that the key bottleneck for the clinical translation of liposomal pMDIs is their low stability. Further, it is regarded that the low stability of the liposomal pMDI system mainly manifests through the aggregation of liposome particles in the propellant medium. Most liposome systems synthesized in previous studies used water as the dispersion medium [53], while the liposomal pMDI system uses propellant. The conventional propellant for a pMDI is chlorofluorocarbons. Nowadays, hydrofluoroalkane (HFA) has replaced chlorofluorocarbons as the main propellant in pMDIs [54]. The main HFA categories used in marketed products are 1,1,1,2-tetrafluoroethane (HFA 134a) and 1,1,1,2,3,3,3-heptafluoropropane (HFA 227). And 1,1-difluoroethane (HFA-152a) is at a promising stage of development. Importantly, all of them are hydrophobic homologs [54,55]. However, the majority of phospholipid material used in liposomes possesses hydrophobic acyl moieties with a high degree of freedom, which is inhibited in an aqueous environment yet increases in a hydrophobic microenvironment. In the hydrophobic propellant medium, the diffusion rate of phospholipids will be significantly higher than that in the aqueous medium. Hence, the bilayer structure of liposomes will become less compact [56], making them more prone to collision-induced particle fusion due to Brownian motion, ultimately leading to the formation of aggregates (Figure 3A).

It should be noted that pulmonary delivery is aimed at enriching therapeutics in the pulmonary region [57] and avoiding mucociliary and macrophage clearance [58]. Monodisperse aerosol particles can improve deposition in the pulmonary region [59]. Additionally, particles with large sizes may be entrapped by lung macrophages [60]. Therefore, the instability-associated aggregation affects the dispersity, increases the clearance rate, and reduces the deposition rate. Ultimately, this can lead to treatment failure and toxic side effects.

Another factor that must be taken into account is the mechanical stress-induced destruction of liposomes (Figure 3B). During the drug delivery process, the HFA propellant generates high vapor pressure to rapidly evaporate the liquid components in the pressure vessel, forming an aerosol [51]. The interface tube is responsible for providing mechanical shear force to break the drug formulation into smaller aerosol droplets [61]. Of note, liposomes are soft lipid nanomaterials with weak mechanical strength [62]. The physical forces can easily destroy the liposome structure, leading to drug leakage, negatively impacting the therapeutic effect. This factor adds to the instability issue of liposomal pMDIs.

In summary, the leading reason for the low stability of liposomal pMDIs lies in the propensity of liposome diffusion, which subsequently results in aggregation, induced by the propellant medium. Furthermore, mechanical forces also contribute to the structural destruction of liposomes. These two destabilizing factors impede the further clinical translation of liposomal pMDI.

## 4. Plausible Strategies for Improving the Stability of Liposomal pMDIs

Breaking through the bottleneck of low stability in liposomal pMDIs is crucial for promoting the clinical translation of related products. With this purpose, continuous efforts have been made, and some plausible strategies have been put forward.

The use of polymer excipients to create a physical barrier is a commonly used method for enhancing the stability of micro–nano systems. This approach seeks to modify the micro–nano systems with polymer excipients, either through non-covalent or covalent interactions, to create a surface barrier that provides steric hindrance, prevents particle aggregation, and resists mechanical forces (Figure 3C) [63]. For instance, studies have been conducted to produce physical barriers using polymer excipients like Tween, Poloxamer, and polyethylene glycol (PEG). These physical barriers have improved the system’s stability to varying degrees (Table 3).

However, it should be noted that the current physical barrier strategy cannot practically address the issue of low stability in liposomal pMDIs. Most studies apply physical barrier strategies to micro–nano systems dispersed in aqueous medium, according to Table 3. It is important to consider this ramification when designing drug delivery systems. The conformation state of polymer excipients differs greatly between aqueous medium and hydrophobic propellant medium [73]. In the aqueous medium, the steric effects of the physical barrier can be assured. Nevertheless, in the propellant medium, the hydrophobic blocks of the polymer excipient molecules fully extend, losing the steric hindrance effect and resulting in a loosened structure. This leads to increases in particle fusion and aggregation, ultimately causing the physical barrier to fail (Figure 3D).

The plausible physical barrier strategy may be useful for aqueous liposomal formulations, but inappropriate for liposomal pMDIs. From this standpoint, effective strategies are urgently required to enhance the system stability of liposomal pMDIs.

## 5. Shell Stabilization Strategy Inspired by Biomineralization

From the above discussion, it can be seen that the instability problems of liposomal pMDIs are mainly the structure-loosened aggregation, as well as the structural damage under mechanical forces. Therefore, it is important to render liposome structures to maintain density and stability even in the hydrophobic propellant medium for further applications.

Biomineralization, to this aim, can be introduced. It is a functional strategy in the process of biological evolution that can make organisms more adaptable to the environment and evolve in a more favorable direction [74]. The formation of natural hard tissues such as eggshells is a classic example of biomineralization. Calcium carbonate is the main component in an eggshell that protects itself from external damage [75,76]. Inspired by the excellent protection properties of the natural biomineralized layers upon organisms under complex environments, we imagine that the stability of liposome particles can be remarkably raised in the propellant medium using similar strategies. The shell stabilization strategy (SSS) (Figure 3E) is thus proposed to drive liposomal pMDIs out of the clinical translation dilemma caused by low stability.

The SSS refers to the layering of appropriate materials on liposomes to form a core–shell nanocomposite with stability in the propellant medium and resistance to mechanical forces. Using liposome as the core template, the mineralized elements and various other protective materials are selectively deposited to form a dense and stable protective layer structure. Depending on the choice of shell material and the formulation method, SSSsare classified into biomineralization, biomineralization-like, and layer-by-layer (LbL) assembly.

Liposomes processed following an SSS are endowed with better shell-derived properties, including improved thermal and chemical stabilities [77]. The elastic properties to resist mechanical forces can also be refined [78]. In addition, the composite structure can provide advantages such as controllable particle size, surface functionality, high drug loading, the inclusion of multiple therapeutic agents, better biocompatibility, and an adjustable release profile [79]. Based on these advantages, the design and synthesis of liposomes using an SSS has attracted great attention and made great progress [80].

The following contents shed light on the three aspects of SSSs, i.e., biomineralization, biomineralization-like, and LbL assembly, as well as offer valuable suggestions for liposomal pMDI development.

### 5.1. Biomineralization

#### 5.1.1. Biomineralization Can Enhance the Structural Integrity and Durability of Biomaterials

In nature, hard tissue materials, such as bones, teeth, shells, etc., are often formed by a specific deposition of inorganic elements on organic substrates with the participation of living cells [81,82,83]. These structures, induced by complex biological systems, often have multi-level ordered structures from the macroscale to the nanoscale [84]. Moreover, they have good mechanical properties and important biological functions. For example, they can help organisms sense signal transductions in the environment, and in particular, they can provide excellent mechanical support and protective functions for organisms [85]. The diatom’s resistance to external pressure or aggression is achieved by forming a biomineralized layer as a surface coating, and the skeletal support of the plankton stems from the cell wall, both of which are mainly composed of silicon [81,86]. In general, cells that cannot form biomineralized layers have a relatively poor survival rate [87]. 

The understanding of biomineralization in nature has promoted the research of biomimetic mineralization. In recent decades, biomimetic mineralization has been applied to many fields, using various biomimetic materials based on the interplay between organic molecules and inorganic elements [88,89]. In the biomedical field, biomineralization is widely used in vaccine improvement, tumor therapy, tissue repair, carrier design, imaging, and the construction of functional living materials [90,91,92] (Figure 4).

Furthermore, the enhancement of material stability through biomineralization typically does not compromise their clinical effectiveness. The nanominerals produced via biomineralization in biological systems have strong dispersion and stability, enabling the precise control of the in vivo behaviors of drugs through biomineralized nano-sized regulation [92]. Fu et al. [93] constructed an inhalable biomineralized liposome coded as LDM co-loaded with dihydroartemisinin (DHA) and pH-responsive calcium phosphate. The in vivo studies showed that LDM nebulization had the highest efficacy in inhibiting tumor growth and resulted in the lowest tumor-associated bioluminescent intensity observed in the lungs. Moreover, speculations on major organs, body weight, and hemograms of tumor-bearing mice indicated the excellent biocompatibility and biosafety of the LDM treatment.

#### 5.1.2. Basic Construction Principles of Biomineralization

In the process of natural biomineralization, the organic phase provides the template and guidance for the construction of the inorganic phase, while the inorganic phase provides mechanical support and protection for the organic phase [94]. The detailed process of biomineralization mainly includes the following. (1) The preassembly of organic macromolecules: proteins, polysaccharides, nucleic acids, and other macromolecules are preassembled to form an ordered reaction environment and determine the location of inorganic nucleation. (2) The mutual recognition of organic–inorganic interface molecules: the nucleation site, crystal phase, crystal type, orientation, and morphology of inorganic materials are controlled by the lattice geometry, electrostatic interaction, polarity, stereochemical complementarity, hydrogen bond interaction, spatial symmetry, and morphology of organic matrix molecules at the interface. (3) Growth regulation: inorganic crystals are assembled in an orderly pattern to form subunits over time, while organic matrix molecules continue to regulate crystal growth. (4) Cell processing: mineralized subunits are further assembled to form a multilevel structure of hard tissue materials [95,96,97,98]. This process is identified in living cell systems. During these four processes, a large number of organic substrates achieve precise assembly, crystal orientation, and structure arrangement from the molecular to macroscopic level by controlling the nucleation, growth, crystal type, and tendency of inorganic crystals [99].

#### 5.1.3. Outstanding Organic Template for Biomineralization: Liposome

It is widely recognized that living cells serve as natural organic templates for biomineralization. Leveraging their excellent interplay, biomineralization has been extensively employed to enhance the stability of living cell storage and delivery [100]. For example, Sun et al. [101] reported an in situ bionic construction strategy with a functional mineral shell. They first mineralized calcium phosphate in situ on the cell wall of the model strain *Acetobacter xylinum*, and in subsequent activity tests, they found that the artificial mineral shell could be an ideal barrier against natural toxins without interfering with normal metabolism.

Liposomes are cellular vesicle analogs that possess a membrane structure and composition similar to those found in living cells [12]. The successful utilization of biomineralization strategies in living cells highlights the potential of liposomes as organic templates. Further investigations revealed that the two primary constituents of liposomes (phospholipids and cholesterol) along with their spherical structure exert a profound regulatory influence on inorganic element deposition.

Phosphorus is the main element in the body that makes up minerals and fats. The majority of the phosphorus is distributed in tissues that are mainly composed of minerals, such as bones and teeth [102]. In fat, phosphorus is mainly found in a type of organic phosphate: phospholipids [103]. High-fat content in the human body is closely related to increased mineralization [104]. Phosphorus compounds can directly form the mineralized material needed, such as apatite or phosphorylation collagen. In addition, they can indirectly regulate the biomineralization process through phosphorus circulation [105]. Among them, the molecular conformation of the phospholipid headgroups, the stereogeometric conformation, and the electrostatic affinity for mineral elements are important factors affecting the interaction [106,107]. In addition, cholesterol is essential for maintaining the fluidity of lipid membranes and promoting the order and rigidity of membrane structures in the fluid state [108]. Thus, phospholipids and cholesterol play an important part in natural biomineralization processes.

Lipid vesicle structures, including liposomes, can be used to isolate metal ions and control ion transport. At the same time, they can control the shape, size, and even orientation of inorganic mineral particles deposited in templates [109]. Stupp et al. [110] found that geometric constraints related to the morphology of nanostructures play a key role in mineralization. The nucleation and growth of hydroxyapatite (HAP) crystals could not be directly controlled using nanostructures with flat surfaces, whilst the nucleation and orientation of HAP crystals could be realized on curved nanostructures (like those possessed by liposomes).

To sum up, a liposome, as a spherical nanoparticle containing phospholipids and cholesterol, has a structure similar to that of a highly ordered cell membrane. The lipid components can regulate the crystallization process of inorganic materials by selectively enriching and localizing inorganic elements [111]. Therefore, using liposomes as templates to induce biomineralization can simulate the mineralized environment in vivo, not only to localize the nucleation and growth of mineral crystals, but also to simulate the guiding role of organic substrates [112].

#### 5.1.4. Calcium-Based Shell

In the previous section, a preliminary statement was made on how the structure and composition of liposomes affect biomineralization. Based on the general recognition that calcium is the most widely distributed element in biomineralized tissues [113], a study on the biomimetic mineralization of calcium compounds on liposomes is considered quite representative and can shed some light on the SSS.

Many calcium-based compounds can be used as inorganic mineralized materials, among which calcium phosphate and calcium carbonate have attracted great interest [114]. Calcium phosphate often exists in the form of different subtypes, commonly including amorphous calcium phosphate (ACP), octacalcium phosphate (OCP, Ca_8_(HPO_4_)_2_(PO_4_)_4_-5H_2_O), calcium hydrogen phosphate dihydrate (DCPD, CaHPO_4_-2H_2_O), calcium-deficient apatite (CaDHA), and HAP, Ca_10_(PO_4_)_6_(OH)_2_) [115,116]. In biomineralized tissues, HAP is the major form of calcium phosphate. OCP is an enamel precursor phase; other calcium phosphates may exist more commonly as precursor phases of HAP. Calcium carbonate is the most abundant biological mineral in nature [117]. Calcium carbonate exists in the form of various polymorphs (calcite, aragonite, and vaterite) and hydrates (monohydrate, hexahydrate, and amorphous (amorphous calcium carbonate, ACC)). Of these, calcite and aragonite are regularly deposited as biominerals [118].

Under the interaction between calcium ions and phospholipids, the phospholipid headgroups undergo a structural change. Additionally, the membrane thickness and particle size of liposomes increase [119]. It is suggested that calcium ion accumulation in the phospholipid bilayer leads to local deformation and stiffness improvement [120].

At present, a large number of liposome-based organic templates have been used to study the formation of mineral layers that regulate calcium carbonate and calcium phosphate. Szcześ et al. [121] used dipalmitoylphosphatidylcholine (DPPC) as a template to study the effect of liposomes on calcium carbonate precipitation. It was shown that calcium ions were attracted to the headgroups of phospholipid, resulting in calcium ion enrichment near the surface of the liposomes and supersaturation in a certain region. At the same time, these enriched calcium ions attracted carbonate ions, which increased the nucleation rate of calcium carbonate. Similarly, zwitterionic or anionic phospholipids such as phosphatidylcholine [122], dioleyl-sn-glycero-3-phosphate (DOPA), and dipalmitoylsn-glycero-3-phosphate (DPPA) [123], polysaccharide-coated dimyristoylphosphatidylcholine (DMPC), and dilauroylphosphatidic acid (DLPA) [124] were also used in the mineralization as organic templates.

When confronted with a plethora of liposome templates, the subsequent factors can be taken into consideration when optimizing the mineralization effect.

Firstly, in the biomineralization process, the charge of the phospholipid headgroups is a crucial factor. Positively charged liposomes show better adsorption efficiency for calcium ions. Smistad et al. [125] found that HAP adsorption characteristics on liposomes are affected by lipid composition (Figure 5A). The charge type is the main factor affecting the adsorption capacity of HAP. In particular, positively charged liposomes show significantly higher adsorption levels than negatively charged liposomes. Further, for positively charged liposomes, the “major lipid type” was found to be an important factor. Erceg et al. [126] used liposomes with different charges in their study: neutral DMPC, negatively charged 1,2-dimyristoylsn-glycero-3-phospho-L-serine (DMPS), and positively charged 1, 2-dioleoyl-SN-glycero3-ethylphosphocholine (EPC) (Figure 5B). Their effects on precipitation, transformation kinetics, and the formation of precipitates in supersaturated aqueous solutions and solid phases of calcium phosphate and calcium carbonate were evaluated. The results showed that the positive DMPS had the most significant effect on the morphology of calcium crystals. This might be owing to a specific electrostatic interaction between phosphatidylserine (PS) and the calcium ions present on the mineral surface.

Secondly, the selection of phospholipid species in liposomes plays an important role in the process of mineralization. Szuki et al. [127] introduced an amphipathic molecule with a bisphosphonate (BP) headgroup that recognized and bonded to HAP. BP is a stable analog of pyrophosphate and has a high affinity for HAP. The content of the BP group was crucial in determining the binding ability of HAP to liposomes (Figure 5C). Arias et al. [128] used gas diffusion to investigate the effects of different lecithin mediums on calcium carbonate crystallization. They also evaluated the effect of the spatial arrangement of lecithin molecules on the formation of template calcium carbonate crystals (Figure 5D). The study demonstrated that the concentration of lecithin in an ionized calcium chloride solution had an impact on the formation of carbonate crystals by influencing the spatial geometry and distribution of lecithin. Therefore, adjusting the assembly of lecithin molecules could lead to alterations in the texture, polymorphism, size, and shape of calcium carbonate crystals. Additionally, research has shown that the combination of phosphatidic acid (PA) or phosphatidylserine (PS) with a lipid membrane could promote the formation of calcium phosphate [126]. However, the combination of calcium with acid phospholipids such as PA, PS, or phosphatidylinositol (PI) would not increase the interaction between the membrane and calcium, or influence the nucleation and growth of calcium carbonate [129]. Lipid components can affect the melting temperature and permeability of liposomes, as well as control precipitation at specific target temperatures through the direct and/or indirect regulation of mineral elements [130].

Thirdly, ionophores can assist in the process. In laboratory preparations, liposome mineralization is typically achieved through either the titration method [131] or by suspending liposomes in simulated body fluid [132]. According to an early study [133], the addition of an ionophore X-537A in a metastable suspension induced calcium carbonate deposition on the surfaces of liposomes rather than inside. Recently, Guo et al. [134] developed an autologous tumor vaccine by engineering a *Salmonella* (Sal) biomineralized with calcium carbonate (Figure 5E). In this study, the facile coating of CaCO_3_ onto the surface of Sal was achieved through co-loading with calcium ionophore A23187. 

Finally, the physicochemical properties of the shell can be influenced by the outer surface properties of the liposome, such as size and curvature. This allows for a more diverse and controllable design of the shell. Smooth calcium carbonate shells can be successfully formed around liposomes with high-curvature surfaces [135]. In some cases, a highly porous calcium carbonate shell could be constructed by tuning the liposome surface properties [136].

**Figure 5 ijms-25-03261-f005:**
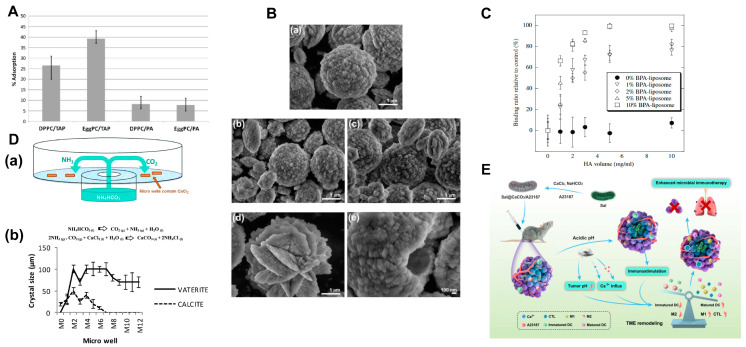
(**A**) Adsorption of charged liposomal formulations to HA. Reproduced from [125]. Copyright 2010, Elsevier. (**B**) SEM micrographs of vaterite precipitated after 10 min of aging time in the (**a**) absence and presence of different liposomes: EPC (**b**), DMPC (**c**), and DMPS (**d**,**e**). Reproduced from [126]. (**C**) Affinity of BPA-liposomes to HA. Reproduced from [127]. Copyright 2009, Elsevier. (**D**) (**a**) A gas diffusion method, (**b**) the types and sizes of crystals obtained in each microwell as a function of lecithin concentration. Reproduced from [128]. Copyright 2015, Royal Society of Chemistry. (**E**) Schematic of the construction of Sal@CaCO_3_/A23187 for enhanced immunotherapy. Reproduced from [134].

#### 5.1.5. Shell Based on Other Material

After comprehending the constructing principles and specific influencing factors for an SSS design involving calcium compounds, it is possible to extend the inorganic materials to other options.

Silicon is a type of metal-like element. Due to its low cost, excellent biocompatibility, and stability, silicon has garnered significant interest in biomineralization [137]. Siliceous fragments in organisms are primarily composed of amorphous silicon dioxide [138]. Mesoporous silica nanoparticles have adjustable porous structures, high specific surface areas, and are easy to functionalize, making them advantageous and widely used for biomedical applications [139]. The composition and structure of a silicon shell can be controlled by adjusting the surface features, such as charge and functional group [140]. The mechanical properties of silica shells can be controlled using different silica precursors with varying chemical structures. For instance, triethoxyvinylsilane (TEVS) [141,142] can be used for a soft shell, while Tetraethoxysilane (TEOS) [143] can be used for a hard one. The deposition of silica on the surfaces of liposomes is structure-dependent on the silanization condition [144] and influenced by the reaction conditions. Chang et al. [142] used TEOS through a sol–gel silicide to endow the liposome with a silicon shell (Figure 6A). This work reported that the formation of solid silica encapsulated liposomes (SLPs) was related to the TEOS concentration, reaction time, temperature, and solvent volume ratio. In addition, this work pointed out that the hydrophobic drug-loading capacity was 2.3 times higher than that of pristine liposomes. This implied that the incorporation of silicon shells into liposomes could be a promising strategy for enhancing both the stability and drug delivery capacity, particularly in the context of hydrophobic drug delivery.

Furthermore, various deposition methods and liposome templates have been developed to regulate silicon shell synthesis more precisely and diversely. Tartis et al. [145] used liposomes as organic templates and adopted the Sol-Generating Chemical Vapor into Liquid (SG-CViL) deposition strategy to control the deposition of silica (Figure 6B). The growth of silica particles was influenced by the composition and concentration of the deposition buffer ions. Moreover, this study revealed that electrostatic interactions facilitate more stable liposome–silica interactions in comparison to hydrogen bonding. Tan et al. [146] produced tubular and spiral liposomes that were used as templates to generate non-spherical silica nanocapsules (Figure 6C). The resulting silica coating’s unique shape could influence the degradation rate and is expected to stabilize the internal contents of the liposomes. Vavia et al. [147] used a modified sol–gel method to coat silica onto the surface of paclitaxel-loaded liposomes (PTX liposils). During a physical stability study, it was discovered that silica coating can enhance the stability of the liposomes and even withstand damage from detergent Triton X-100. (Figure 6D). In addition to enhancing physical stability, in vivo pharmacokinetic studies conducted on rats demonstrated the significant potential of PTX liposils in maintaining prolonged drug plasma concentrations. Furthermore, hemolysis studies exhibited their exceptional compatibility with blood components. Hence, it can be concluded that a silica coating effectively eliminates inherent instability issues associated with conventional liposomes without compromising their clinical efficacy.

Moreover, some metallic elements other than calcium are often used in the formation of artificial biomineral shells. Zhang et al. [148] encapsulated the hematoporphyrin monomethyl ether (HMME) and acriflavine (ACF) in liposomes. They later coated ultra-thin manganese dioxide (MnO_2_) nanosheets on the liposome surface using a REDOX reaction with potassium permanganate and PEG. The tumor-targeting AS1411 aptamer was conjugated to obtain the Lipo/HMME/ACF@MnO2-AS1411 delivery system. The in vivo studies demonstrated that this delivery system exhibited prolonged blood circulation characteristics and enhanced efficacy in tumor therapy. Interestingly, the presence of manganese dioxide nanosheets also endowed the drug delivery system with a great potential in tumor real-time magnetic resonance imaging applications (Figure 6E). The shell structure of the magnetic material not only enhanced stability, but also facilitated the integration of liposomes into a comprehensive drug delivery system for theranostics, thereby presenting a promising application for liposomes. In addition to monotonic metal biomineralization, several studies have reported the use of hybrid metal complexes. Liu et al. [149] investigated the possibility of lanthanide ions in surface sediments induced by liposomes. Lipids containing PS species were utilized to form a gadolinium/adenosine monophosphate (Gd^3+^/AMP) shell on liposomes (Figure 6F). The liposomes were effectively shielded by the shell, preventing drug leakage in the presence of Triton X-10 and ZnO nanoparticles.

#### 5.1.6. Surface Modification

The lipid surface of the liposome can be modified with functional macromolecules, such as proteins, DNA, enzymes, and polysaccharides, which subsequently undergo ion-induced biomineralization, to create a system that closely resembles naturally existing stromal cell vesicles. This modification allows for the formation of a stable mineralized shell. Sternik et al. [150] used Phospholipase A2 (PLA2)-modified liposomes as organic substrates. PLA2 catalyzed the hydrolysis of phospholipids, producing fatty acids and lysophospholipids. This affects the surface free energy and negative zeta potential of the lipid layer, which in turn affects the deposition of inorganic elements on its surface. Ciancaglini et al. [151] revealed the new role of Annexin A5 in calcification. The study investigated the ability of annexin A5 to adsorb on matrix–vesicle biomimetic liposomes and Langmuir monolayers in the absence and presence of calcium ions. It was shown that A5 in matrix vesicles was recruited to membrane sites enriched in PS and calcium ions. This recruitment occurred not only to contribute to intraluminal mineral formation, but also to stabilize the vesicle membrane and prevent premature rupture. Proteoglycans are also potential surface modifiers of liposomes, bearing a complex spatial structure and different charge densities, which directly or indirectly participate in the biomineralization process [152]. Electronic interaction plays a crucial role in the biomineralization process [152]. Calcium ions often bind to negatively charged organic molecules. It is also possible to modify the surfaces of liposomes with charged molecules. For instance, the structure of bovine serum albumin (BSA) contains reductant disulfide bonds, thiol groups with modification potential, and numerous charged side chain acids that can offer ample binding sites for liposomes [153]. Similarly, due to their high negative charge density, DNA molecules are also suitable for guiding biological mineralization [154].

Based on the aforementioned discussion, it is evident that biomineralization offers a straightforward and cost-effective nano-modification approach for diverse drug-carrying liposomes. Biomineralized materials derived from various sources enable the encapsulation of liposomes within diverse protective shells. Investigating the interaction between mineralized elements and liposomes enhances the efficacy and controllability of this strategy by uncovering crucial influencing factors. Moreover, biomineralization demonstrates its capability to confer remarkable resilience upon liposomes against adverse external conditions such as detergent, enzymes, and mechanical forces. Consequently, the core–shell structure formed by employing biomineralization holds promise in addressing the issue of the low stability of liposomal pMDIs. Additionally, this artificial, mineralized shell exhibits excellent biodegradability, biocompatibility, and physiological stability in drug delivery to the human body [92].

### 5.2. Biomineralization-like

Proteins, in addition to the inorganic mineral materials mentioned above, can be employed in an SSS to construct shell structures that support and protect liposomes. As this process is similar to biomineralization, we will define it as a biomineralization-like process hereby.

#### 5.2.1. Artificial Protein Corona

Liposomes’ high specific surface area provides good surf-activity, allowing proteins to adsorb to their surface in natural or denatured forms [155,156]. This adsorbed protein layer can alter the size, morphology, and stability of liposomes [157]. This protein layer is commonly known as the ‘protein corona’ (PC) and has been extensively studied. Although research on the PC is more focused on targeting, cellular uptake, and safety [158,159], importantly, some types of proteins can assemble into rigid, crystalline, and functional PC structures [160,161]. Moreover, some proteins own specific physiological and pharmacological functions [162]. Therefore, it is reasonable to speculate that synthesizing an artificial engineering PC on liposomes could be a potential alternative for the SSS.

The challenge of this strategy lies in adsorbing a stable homogeneous protein layer onto the liposome surface. The interaction mechanism between proteins and liposomes is complex and influenced by various factors, such as size [163], charge [164], shape [165], surface roughness [166], ligand structure, and steric hindrance [167]. These factors affect the number and category of adsorbed proteins during PC formation. In general, nanoparticles that are positively charged, non-spherical, large in size, and have a smooth surface interact more intensively with proteins [168]. Specifically, for liposomes, the surface charge is a crucial factor [169]. Caracciolo et al. [170] used cationic, neutrally charged, and anionic liposomes to investigate PC formation. The amount of protein adsorbed on the surface of positively charged liposomes was significantly higher than that of the other two counterparts.

#### 5.2.2. Potential Transformation of Protein Corona

Noticeably, the polar groups on the polypeptide chains of proteins readily adsorb water molecules, leading to the formation of a hydration film surrounding the protein particles. This hydration film plays a crucial role in maintaining protein colloid stability [171]. However, upon the formation of an artificial PC on the liposome surface, it is possible that the propellant medium could disrupt the hydration layer between protein particles. Furthermore, considering that proteins and liposomes share similar surface hydrophilic and internal hydrophobic structures [172], it can be hypothesized that they would exhibit comparable diffusion characteristics resulting in collision deposition. These two factors imply that there is a possibility of encasing proteins in the outer layer of liposomes through deposition rather than adsorption. Indeed, studies have documented the formation of stable protein particles through precipitation, which can be utilized for the effective encapsulation of the contents. McClements et al. [173] produced core–shell protein nanoparticles through antisolvent precipitation using a continuous dual-channel microfluidization method. The solvent phase (zein in ethanol) and antisolvent phase (casein in water) were used to prepare small core–shell protein nanoparticles with a diameter of 125 nm. Thus, the further denaturing deposition of the PC adsorbed on liposomes can be considered. The proteins will be transformed into a rigid and dense polymeric layer after microcoagulation, triggered by solvent–antisolvent shift. However, the mechanism and practical possibility of protein aggregation and deposition on liposomes through this approach still need to be further explored.

In general, a biomineralization-like strategy enhances the understanding and knowledge of mineralized materials, thereby enabling diverse designs for the protective shell. However, it is important to note that current research on biomimetic single-shell structures is primarily focused on polymer studies [174,175,176], rather than artificial PCs. Nevertheless, as discussed in the previous section, addressing the urgent issue of maintaining the effective barrier function of polymers in the hydrophobic medium remains a challenge. Taking inspiration from artificial PCs, scientists in the drug delivery field should explore alternative protein materials and seek to enhance material stability for enhancing liposomal pMDI stability.

### 5.3. LbL Assembly

Among the biomimetic mineralization approaches mentioned above, a single layer is constructed on the liposomes. Noticeably, in a hydrophobic medium, the monolayer shell structure may function as an inefficient physical barrier in terms of enhancing the stability of liposomal pMDIs. [73]. An LbL assembly strategy can be employed to sequentially stabilize liposomes by incorporating mineral materials and other functional materials such as proteins and polymers into a multilayer shell structure. Some polymeric materials and proteins themselves can provide active sites for biomineralization, thus improving the biomineralization ability of liposomes [177]. From this viewpoint, the LbL assembly strategy has fewer restrictions on the materials. LbL assembly enables facile coating with a range of physicochemical properties and geometers without damaging the substrate [178]. Moreover, the layer formed through LbL assembly may provide higher degrees of mechanical strength and protection than a single layer [179]. At the same time, LbL assembly possesses a simple and controllable assembly process, and a superior regulation function, which becomes an effective tool for surface modification during biomedical material design [180,181].

Protein materials are one of the most commonly used materials for LbL assembly. Zhao et al. [182] utilized this approach. Protein nano gels embedded with porch pancreas lipase (PPL) were first reacted with BSA. Once the BSA molecules were uniformly coated on the surfaces of the nanogels, the bionic mineralized shell was formed on the BSA layer using calcium carbonate, and the nanogels aggregated into suprastructures (Figure 7A). Subsequent experiments showed that the LbL structure could effectively maintain the biological activity of PPL molecules in the presence of trypsin.

LbL self-assembly strategies have been developed for a wide range of polymeric materials, such as polyelectrolyte materials [183]. This allows for a broader scope of metal element deposition regulation, extending the existing mechanisms to polyelectrolyte interactions. Polyelectrolyte materials are typically classified as natural (e.g., gelatin, lysozymes, albumin, nucleic acids, and polysaccharides) and synthetic (e.g., poly(styrene sulfonate) (PSS), poly(allylamine) (PAH), poly-l-lysine (PLL), poly (dimethyl diallyl ammonium chloride) (PDDA), poly(ethylenimine) (PEI), poly(N-isopropyl acrylamide)(PNIPAM), poly(acrylic acid) (PAA), poly (methacrylic acid) (PMA), and poly(vinyl sulfate) (PVS)) [184]. It can also be divided into polycations (e.g., chitosan, polyethyleneimine hydrochloride, polyvinylpyridine, polyvinylamine) and polyanions (e.g., sodium alginate, sodium polyacrylate, polystyrene sulfate, polyethylene sulfonic acid, polyethylene phosphate) based on the charge of the ions present.

In the LbL assembly strategy, polycationic materials can be deposited on the surfaces of liposomes to form a transition layer. The polyanions can be subsequently deposited onto the polycations layer, forming multi-layer shell structures through electrostatic and nonelectrostatic interactions [185]. Electrostatic interactions serve as the predominant driving force in LbL assembly. Nonelectrostatic interactions include van der Waals, hydrophobic, hydrogen bond, host–guest, coordination bond, and other forces [186]. The multiple thin shells can enhance the net charge and active site on the surfaces of liposomes [187]. Then, redeposition promotes the deposition of functional shells. In this process, the physical structural properties of the shell can be controlled by adjusting factors such as pH, time, temperature, ionic strength, coating material concentration, and washing and drying conditions [188,189,190].

By manipulating the aforementioned regulatory factors, the polyelectrolyte interaction can be conveniently modulated to tailor the stiffness of the LbL films [179]. Furthermore, the mechanical properties of LbL films are largely dictated by the composition of layer materials [191]. Hence, employing materials with suitable strength and hardness enables an expansion in the stiffness range of the original film (kPa to MPa) toward higher levels (GPa) [192]. Tang et al. [193] studied *Saccharomyces cerevisiae* (yeast) cells that could not spontaneously synthesize a mineralized layer with a similar structure to the liposome membrane (Figure 7B). They used poly (diallyldimethylammonium chloride) (PDADMAC) and sodium polyacrylate to repeatedly deposit on the surfaces of the cells. Sodium polyacrylate provided a high density of carboxylate groups as a calcium ion deposition site. The synthetic mineral shell, created in situ, provided protection against lytic enzymes and extended the storage time of yeast cells. Fujimoto et al. [124] placed phosphate ions and calcium ions in the inner cavity and outer phase of liposomes, respectively. The liposome surface was coated with polysaccharide compounds, specifically chitosan (CHI) and dextran sulfate (DXS), using an LbL assembly strategy. The polymerization layer served as both the reaction site for calcium phosphate deposition and the regulator of ion diffusion. By coating liposome surfaces with various polysaccharides and adjusting reaction conditions, it was possible to control the formation, deposition sites, and crystal properties of calcium phosphate (Figure 7C). Song et al. [194] sequentially deposited CHI and pea protein isolate hydrolysates (PPIHs) onto flaxseed oil liposomes (FL Lipos), resulting in the formation of a double-shell encapsulated liposome (FL Lipo-C-P). Notably, upon coating with the second layer of PPIH, electrostatic interactions, hydrogen bonding, and hydrophobic interactions caused the rearrangement of chitosan chains on the liposome surface (Figure 7D). This led to reduced membrane permeability and a denser membrane structure. Compared to FL Lipo, FL Lipo-C-P exhibited enhanced oxidation stability during storage.

**Figure 7 ijms-25-03261-f007:**
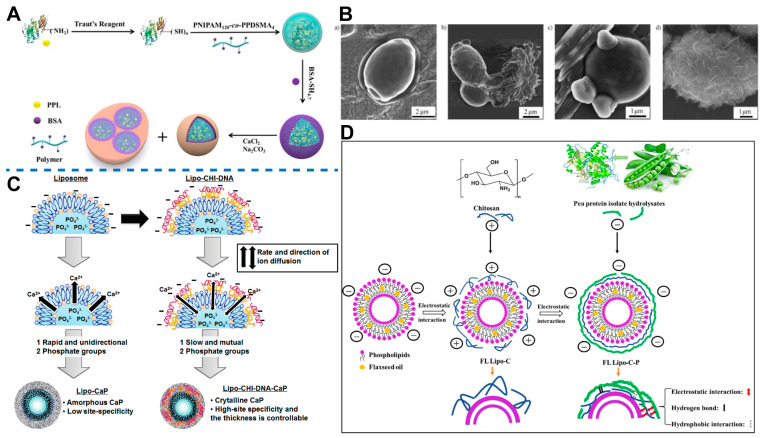
(**A**) Schematic depiction of the synthesis and biomineralization of PPL nanogels. Reproduced from [182]. Copyright: 2019, John Wiley and Sons. (**B**) SEM images: (**a**) Bare *S. cerevisiae*; (**b**) *S. cerevisiae* cells are hardly calcified and calcium minerals precipitate separately; (**c**) some calcium minerals precipitate randomly on the bare cell; (**d**) *S. cerevisiae* with a mineral coat after the LbL treatment. Reproduced from [193]. Copyright: 2008, John Wiley and Sons. (**C**) Schematic representation of controlling mineralization over the surface of nanocapsules by tuning the rate and direction of ion diffusion, surface functional groups, and the reaction conditions Reproduced from [124]. (**D**) Schematic of the formation and stability mechanisms of FL Lipo-C-P. Reproduced from [194]. Copyright: 2023, Elsevier.

In summary, LbL assembly offers a versatile and adjustable approach for fabricating multiple multilayer shells with controlled composition and structure. Through LbL assembly, materials of varying sizes such as small molecules, proteins, polymers, and cells can be combined synergistically to overcome the limitations of individual materials, thereby enhancing the mechanical strength and diversifying the functionality of liposomal pMDIs. Moreover, LbL assembly enables a better emulation of in vivo microenvironments by providing cargo protection, precise targeting capabilities, stimulation-responsive delivery mechanisms, as well as an improved co-delivery of multiple therapies in both the temporal and spatial domains [195].

## 6. Outlook

When selecting stabilization strategies, it is important to prevent selecting metal materials that are toxic to the human body. Meanwhile, efforts should be made to optimize the synthesis strategy and biocompatibility of inorganic materials. For instance, gadolinium ions can lead to severe renal fibrosis and brain damage during long-term use [196], which also explains why it is being replaced by safer iron-based nanoparticles in the market [197]. From this perspective, iron and calcium may be better choices. Additionally, when certain materials interact with liposomes, it is important to consider potential safety issues that may arise from the combination. For instance, liposomes can affect the self-organization of proteins and promote the induction of amyloidosis [198]. This means that the conformation of proteins may change after binding to liposomes, resulting in the exposure of antigenic determinants. Therefore, during the clinical translation process of a liposomal pMDI, it is crucial to simultaneously address the two major issues of stability plus safety.

Currently, research on SSSs is mostly limited to the laboratory scale. Still, many mechanisms and factors influencing shell formation require further study [92], not to mention industrialization. Expanding production from small-scale batches to industrial levels is a significant challenge that requires careful consideration and exploration. As established in this work, the stability of liposomal pMDIs can be enhanced using SSSs. In combination with SSSs, other auxiliary methods can also be adopted, as discussed as follows.

Firstly, optimizing the charge properties of liposomes is supposed to be of help. The instability of liposomes in pMDI systems is caused by their collision deposition. According to Derjaguin–Landau–Verwey–Overbeek (DLVO) theory, the surfaces of charged nanoparticles (e.g., liposomes) attract ions and counter-ions, imposing electrostatic repulsive interactions, which are responsible for maintaining stability. However, uncharged nanoparticles tend to aggregate due to strong van der Waals attraction [199]. Therefore, using charged liposomes to create repulsions may be an additional way to resist and prevent particle collision.

Secondly, by adjusting the formulation composition, the mechanical properties of the liposome bilayer can be improved [200]. It is advisable, for example, to use sphingomyelin, which interacts more intensively with cholesterol than phosphatidylcholine. Also, through intermolecular hydrogen bonds, the sheath formed by sphingomyelin and cholesterol has a more compressible nature than that of conventional liposome. Penetration enhancers (PEs) have been reported to affect the deformability of liposomes. Liposomes with PEs have better elastic properties than traditional ones [201].

Thirdly, appropriate excipients can also be added to the pMDI formulation to exert new functions. Surfactants or co-solvents optimize the dispersibility of liposomes or alter the vapor pressure of the propellant, also contributing to the stability of the pMDI system. The addition of surf-active sodium carboxymethyl cellulose prevents the aggregation of lipid-based nanoparticles in an HFA medium, improving the stability of the system [202]. It was reported that adding 1H,1H,2H,2H-perfluorooctan-1-OL (PFOH) as a co-solvent based on phospholipid formulation can reduce particle adhesion to the tank wall and inhibit particle flocculation [203]. Thus, it can produce pMDI formulation with better physical stability.

From an industrial conversion standpoint, these auxiliary strategies offer low costs, simple operation, and feasible scale-up. Therefore, we believe that combining these approaches with an SSS is believed to aid in the transition of liposomal pMDIs from bench to bedside (B2B).

## 7. Conclusions

Under the background of the gradual increase in lung diseases worldwide and the understanding of numerous therapeutic advantages, liposomes have been developed as an excellent drug delivery vehicle in conjunction with a variety of inhalers. Among these inhalers, pMDIs have won attention with favorable cost performances, patient compliance, and accurate dosages. However, the clinical application of liposomal pMDIs has been hindered by the tendency of the collision-induced aggregation of the liposome lipid bilayer in a hydrophobic propellant medium and brittleness under high mechanical forces. In nature, biomineralization produces hard tissues with highly ordered structures and good mechanical properties, providing effective protection for fragile biological tissues. The process of surface modification through biomineralization is highly biocompatible, simple to operate, economical, and efficient. Inspired by biomineralization, we proposed an SSS to surmount the poor stability of liposomal pMDIs. The organic–inorganic hybrid shell structure formed by the SSS can provide liposomes with excellent physical stability and mechanical properties. Studies on utilizing SSSs in liposomes and similar nanostructures have provided valuable information. These include papers about biomineralization, biomineralization-like, and LbL assembly strategies that can be used as references, where synthesis strategies, factors affecting shell formation, and the protective capacity of the shell were summarized. SSSs can provide new opportunities for the application of liposomal pMDIs. The proposed strategy could also inspire researchers to overcome the difficulties posed by the instability of liposomes in other drug delivery systems.

## Figures and Tables

**Figure 1 ijms-25-03261-f001:**
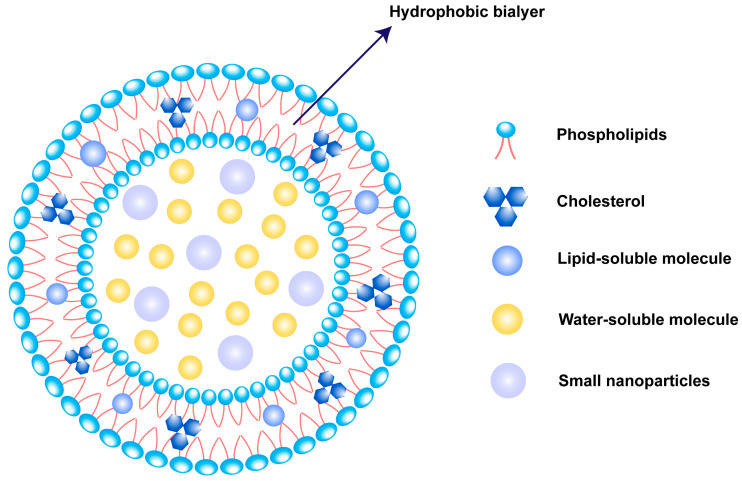
Illustration of liposome nano architectonics.

**Figure 2 ijms-25-03261-f002:**
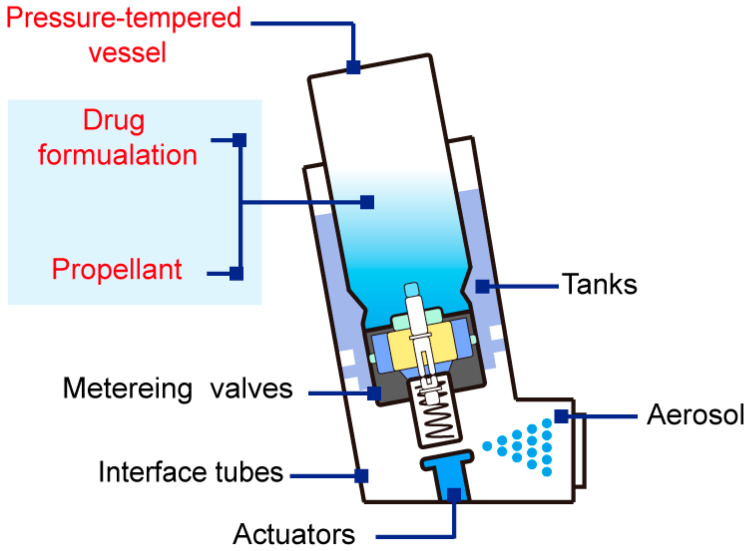
Illustration of pMDI structure.

**Figure 3 ijms-25-03261-f003:**
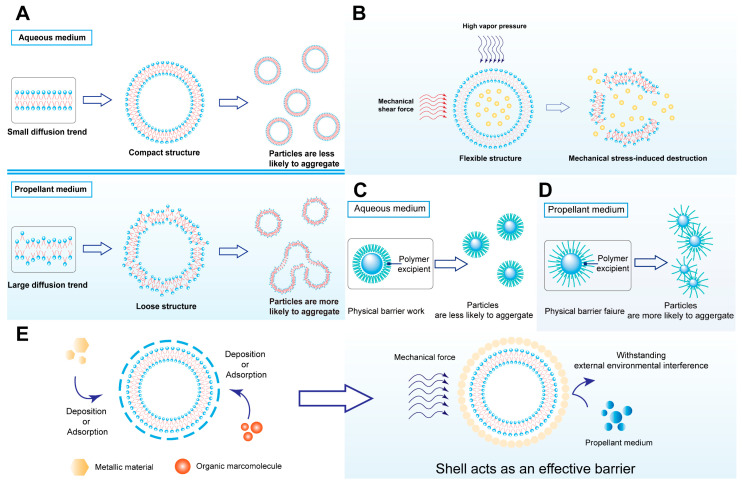
(**A**) Propellant medium-induced low stability of liposomal pMDI; (**B**) Mechanical stress-induced low stability of liposomal pMDI; (**C**) Physical barrier strategy; (**D**) Reason for failure of physical barrier strategy in propellant medium; (**E**) Shell stabilization strategy.

**Figure 4 ijms-25-03261-f004:**
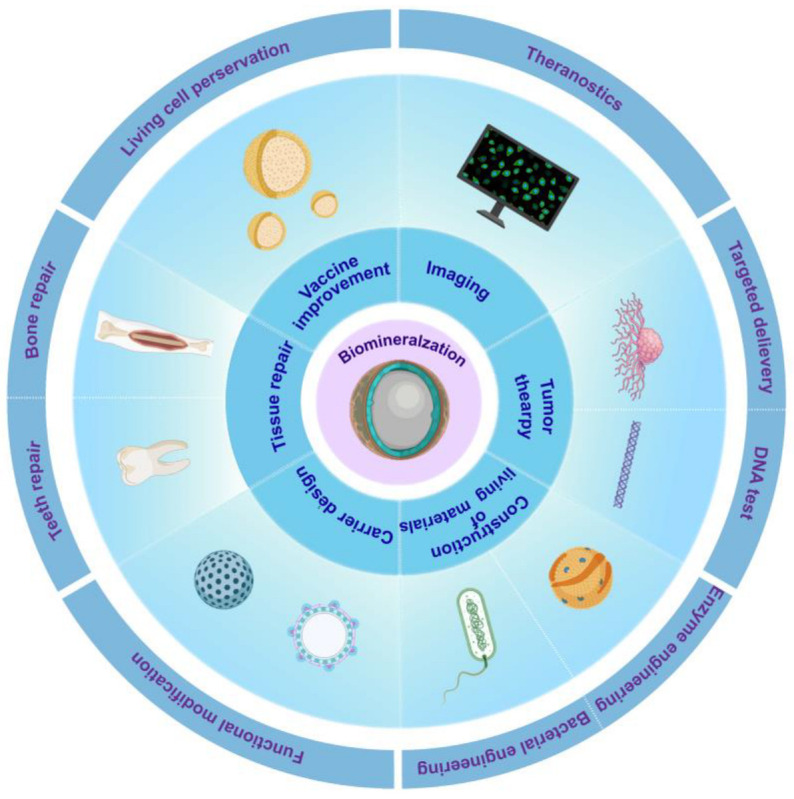
Biomineralization in biomedical applications.

**Figure 6 ijms-25-03261-f006:**
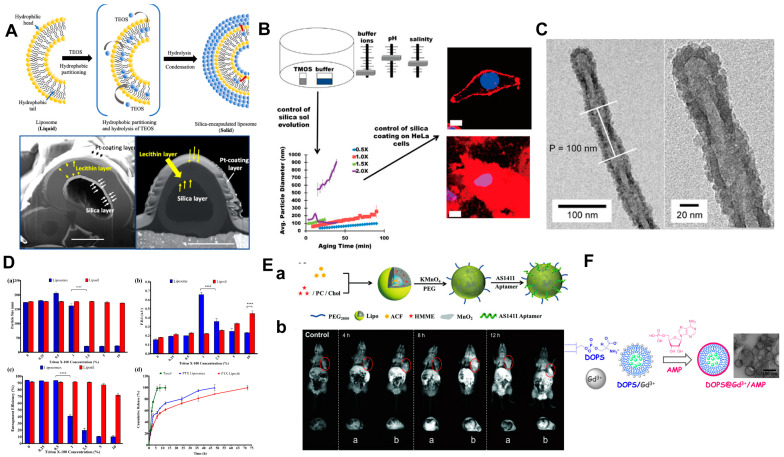
(**A**) Schematic representations of silica encapsulation in a liquid liposome and FIB-SEM images of SLPs. Reproduced from [142]. Copyright: 2021, American Chemical Society. (**B**) Schematic representations of silica sol generation. By varying the buffer ionic constituents, concentration, pH, and sol aging temperature, the silica particle size in silica sols can be controlled. Reproduced from [145]. Copyright: 2017, American Chemical Society (**C**) TEM image of helical silica. Reproduced from [146]. Copyright: 2009, American Chemical Society. (**D**) Effect of Triton X-100 on the physical stability of PTX liposomes and liposils with respect to (**a**) particle size, (**b**) polydispersity index (PDI), (**c**) entrapment efficiency (**** *p* < 0.0001), and the corresponding effect seen from the (**d**) in vitro release profile of PTX from Taxol^®^ (A commercial PTX product.), PTX liposomes, and PTX liposils over a period of 72 h. Reproduced from [147]. Copyright: 2018, Elsevier B.V. (**E**) **a**. Scheme of manganese dioxide-coated liposomes. **b**. In vivo T1-weighted MRI imagesof a Lipo/HMME/ACF@MnO_2_, and b Lipo/HMME/ACF@MnO_2_-AS1411. Reproduced from [148]. Copyright: 2018, John Wiley and Sons. (**F**) Illustration of liposome@Gd^3+^/AMP. Reproduced from [149]. Copyright: 2019, American Chemical Society.

**Table 1 ijms-25-03261-t001:** FDA-approved nanomedicines reproduced from Mitchell et al. [7] with permission from the Asian Journal of Pharmaceutical Sciences.

Drug	Company	Application	Date of First Approved	Types
Injectafer	American Regent, Shirley, NY, USA	Iron-deficient anemia	2013	Inorganic
Feraheme	AMAG, Zurich, Switzerland	Iron deficiency in chronic kidney disease	2009	
Venofer	American Regent	Iron deficiency in chronic kidney disease	2000	
Ferrlecit	Sanofi, Paris, France	Iron deficiency in chronic kidney disease	1999	
DexFerrum	American Regent	Iron-deficient anemia	1996	
INFeD	Allergan, Dublin, Ireland	Iron-deficient anemia	1992	
ADYNOVATE	Takeda, Tokyo, Japan	Hemophilia	2015	Polymer-based
Plegridy	Biogen, Cambridge, MA, USA	Multiple sclerosis	2014	
Cimiza	UCB, Brussels, Belgium	Crohn’s disease, rheumatoid arthritis, psoriatic arthritis, ankylosing spondylitis	2008	
Abraxane	Celgene, Summit, NJ, USA	Lung cancer, metastatic breast cancer, metastatic pancreatic cancer	2005	
Neulasta	Amgen, Thousand Oaks, CA, USA	Neutropenia, chemotherapy induced	2002	
Eligard	Tolmar, Fort Collins, CO, USA	Prostate cancer	2002	
PegIntron	Merck, Rahway, NJ, USA	Hepatitis C infection	2001	
Copaxone	Teva, Tel Aviv, Israel	Multiple sclerosis	1996	
Oncaspar	Servier Pharmaceuticals, Boston, MA, USA	Acute lymphoblastic leukemia	1994	
Onpattro	Alnylam Pharmaceuticals, Cambridge, MA, USA	Transthyretin-mediated amyloidosis	2018	Lipid-based
Vyxeos	Jazz Pharmaceuticals, Dublin, Ireland	Acute myeloid leukemia	2017	
Onivyde	Ipsen, Paris, France	Metastatic pancreatic cancer	2015	
Visudyne	Bausch and Lomb, Laval, Canada	Wet age-related macular degeneration, myopia, ocular histoplasmosis	2000	
AmBisome	Gilead Sciences, Foster City, CA, USA	Fungal/protozoal infections	1997	
DaunoXome	Galen, Craigavon, Northern Ireland	Kaposi’s sarcoma	1996	
Doxil	Janssen, Beerse, Belgium	Kaposi’s sarcoma, ovarian cancer, multiple myeloma	1995	

**Table 2 ijms-25-03261-t002:** Liposome products on the market, reproduced from He et al. [17] with permission from the Asian Journal of Pharmaceutical Sciences.

Product Name	Approval Year	Company	Active Ingredient	Administration Route	Formulation	Indication
Ambisome^®^	1990	Astellas (Chuo-Ku, Tokyo, Japan)	Amphotericin B	intravenous	HSPC, DSPG, and Cholesterol	Fungal infection
Epaxal^®^	1993	Crucell Italy S.r.l. (Baranzate, Italy)	Inactivated hepatitis A virus (strain RGSB)	intramuscular	DOPC and DOPE	Hepatitis A
Abelcet^®^	1995	Leadiant Bioscience Inc. (Gaithersburg, MD, USA)	Amphotericin B	intravenous	DMPC and DMPG	Invasive severe fungal infections
Doxil^®^/Caelyx^®^	1995/1996	Baxter Healthcare Corp. (Deerfield, IL, USA)/Baxter Holding B.V.(Utrecht, The Netherlands)	Doxorubicin	intravenous	HSPC, Cholesterol, and PEG 2000-DSPE	Ovarian cancer and Kaposi’s sarcoma
Amphotec^®^	1996	Alkopharma USA (Sacramento, CA, USA)	Amphotericin B	intravenous	Cholesteryl sulphate	Severe fungal infections
DaunoXome^®^	1996	Galen (Northern Ireland, UK)	Daunorubicin	intravenous	DSPC and Cholesterol	Kaposi’s sarcoma infected with human immunodeficiency virus
Inflexal^®^ V	1997	Crucell Berna Biotech (Bern, Switzerland)	Inactivated hemagglutinin of Influenza virus strains A and B	intramuscular	DOPC and DOPE	Influenza
Depocyt^®^	1999	Pacira Pharmaceuticals Inc (San Diego, CA, USA)	Cytarabine	spinal	DOPC, DPPG, Cholesterol, and Triolein	Neoplastic meningitis
Visudyne^®^	2000	V Valeant Luxem Bourg (Quebec, Canada)	Verteporfin	intravenous	DMPC and EPG	Choroidal neovascularisation
Myocet^®^	2001	Teva B.V. (City of Arkansas, Palestine)	Doxorubicin	intravenous	Lecithin and Cholesterol	Combination therapy with cyclophosphamide in metastatic breast cancer
Lipusu^®^	2003	Luye Pharma Group (Yantai, Shandong, China)	Paclitaxel	intravenously guttae	Lecithin, Cholesterol, Threonine, and Glucose	Ovarian cancer
DepoDur^TM^ Epidural	2004	Pacira Pharmaceuticals Inc (San Diego, CA, USA)	Morphine	sulfate	DOPC, DPPG, Cholesterol, and Triolein	Pain management
Mepact^®^	2009	Takeda France SAS (Paris, France)	Mifamurtide	intravenous	DOPS and POPC	Non-metastatic osteosarcoma
Exparel^®^	2011	Pacira Pharmaceuticals Inc (San Diego, CA, USA)	Bupivacaine	intravenous	DEPC, DPPG, Cholesterol, and Tricaprylin	Pain management
Marqibo^®^	2012	Acrotech Biopharma Inc (East Windsor, NJ, USA)	Vincristine	intravenous	Sphingomyelin and Cholesterol	Acute lymphoblastic leukemia
Onivyde^TM^	2015	Ipsen S.A (Boulogne-Billancourt, France)	Irinotecan	intravenous	DSPC, mPEG-2000, and DSPE	Metastatic pancreatic cancer
Vyxeos^®^	2017	Celator Pharms (Princeton, NJ, USA)	Daunorubicin and Cytarabine	intravenous	DSPC, DSPG, and Cholesterol	Acute myeloid leukemia with myelodysplasia-related changes and therapy-related acute myeloid leukemia
Shingrix^®^	2017	GlaxoSmithkline Biologicals SA (London, UK)	Recombinant VZV glycoprotein E	intramuscular	DOPC and Chol	Against shingles and post-herpetic neuralgia
Onpattro^TM^	2018	Alnylam Pharmas.Inc (Cambridge, MA, USA)	siRNA	intravenous	DLin-MC3-DMA, DSPC, Cholesterol, and PEG2000-C-DMG	Polyneuropathy caused by hereditary transthyretin familial amyloidosis
Arikayce^®^ Kit	2018	Insmed Inc (Bridgewater, NJ, USA)	Amikacin	inhalation administration	DPPC and CHO-HP	Non-tuberculous mycobacteria lung disease caused by mycobacterium avium complex
Comirnaty^®^	2021	BioNTech Manufacturing GmbH (Mainz, Germany)	BNT162b2	intramuscular	ALC-0315, ALC-0159, DSPC, and Cholesterol	COVID-19
Moderna SM-102	2021	Moderna (Cambridge, MA, USA)	mRNA-1273	intramuscular	PEG2000-DMG, Cholesterol, and DSPC	COVID-19

Abbreviations: HSPC: Hydrogenated Soybean Phosphotidylcholine; DSPG: Distearoylphosphatidylglycerol; DOPC: Dioleoylphosphatidylcholine; DOPE: Dioleoylphosphoethanolamine; DMPC: 1,2-Dimyristoyl-sn-glycero-3-phosphocholine; DMPG: 1,2-Dimyristoyl-sn-glycero-3-phosphorylglycerol sodium salt; PEG: Polyethylene glycol; DSPC: Distearoylphosphatidylcholine; DPPG: Dipalmitoylphosphatidylglycerol; EPG: Esterified propoxylated glycerol; DOPS: Dioleoylphosphatidylserine; DEPC: Dierucoylphosphatidylcholine; DSPE: Distearoylphosphoethanolamine; DPPC: Dipalmitoylphosphatidylcholine DMA: Distearoyl Phosphoethanolamine; DMG: Dimethylglycine.

**Table 3 ijms-25-03261-t003:** Cases focused on the implementation of physical barrier strategies utilizing polymer excipients.

Micro–Nano System	Polymer Excipient	Modification Method	Stable Period (Day)	Dispersion Medium	References
Polylactic-acetic acid copolymer nanoparticles	Vitamin E PEG Succinate	Non-covalent binding	90	Water	[64]
Zein nanoparticles	Polysorbate 80	Non-covalent binding	30	Water	[65]
Selenate nanoparticles	Dextran T-70	Non-covalent binding	30	Water	[66]
Tannic acid derivative nanoparticles	Poloxamer 188	Non-covalent binding	2	Water	[67]
Glutaraldehyde crosslinked polyvinyl alcohol microspheres	Span 80	Non-covalent binding	15	Water	[68]
Glutaraldehyde crosslinked polyvinyl alcohol microspheres	Polyvinyl alcohol	Non-covalent binding	14	Water	[69]
Iron oxide–manganese oxide nanoparticles	PEG3000	Covalent binding	90	Water	[70]
Silver nanoparticles	Aminophenylboric acid-polyvinyl alcohol copolymer	Covalent binding	70	Water	[71]
Illite nanoparticles	Polyvinylpyrrolidone (PVP) K10	Covalent binding	14	Water	[72]

## Data Availability

No new data were created or analyzed in this study. Data sharing is not applicable to this article.

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
