# Peer review of "Overcoming the Low-Stability Bottleneck in the Clinical Translation of Liposomal Pressurized Metered-Dose Inhalers: A Shell Stabilization Strategy Inspired by Biomineralization"

_ijms, 2024, doi:10.3390/ijms25063261_

Round 1

Reviewer 1 Report

Comments and Suggestions for Authors

The manuscript is an interesting study about the potential use of shell stabilization strategies such as biomineralization, biomineralization-like, and layer-by-layer (LbL) assembly to solve the problem of low stability of liposomal pMDI. The topic of the manuscript is of great interest and well fits with the aim and scope of IJMS. This review gives people a certain understanding of liposome and liposome pulmonary delivery and provides useful information for beginners in the field. To make the review more comprehensive, it is recommended to make the following modifications:

1.      I think most of the readers know about liposome or pMDI structures. The authors should shorten these sections. The manuscript should pay more attention to focus on the in vivo studies and applications. Explain more and expand the result of others' work in Section 5.

2.      Besides schematic illustration, to help readers better understand the described content, more meaningful figures, key diagrams, and tables, especially in section 5 which is the main part of the manuscript should be provided.

3.      Two Figures 5 were placed and should be corrected. Figures 3,4,5,5, can be combined and replaced by better and higher resolution ones.

4.      Some of the references are too old and should be updated.

Comments on the Quality of English Language

Minor editing of English language required.

Reviewer 2 Report

Comments and Suggestions for Authors

This work describes a shell stabilization strategy to solve the problem of low stability of liposomal pressurized metered dose inhalers. It also covers synthesis strategies and the modification with various molecules/functional moieties. The authors correctly planned the experiments and clearly presented their results. They used a wide range of analytical methods to characterise the resulting material. The depth of the analyses is evidenced by the authors' study of over 200 scientific articles, which they included in the references. Thanks to the solution proposed by the authors, it will be possible to avoid the adverse effect of collision-induced aggregation of the liposome lipid bilayer in a hydrophobic propellant medium and the liposomes instability in various drug delivery systems. In my opinion, the work can be published in Int. J. Mol. Sci. In the present form.

Comments on the Quality of English Language

English language requires minor corrections.
